# Biomechanics of keratoconus: Two numerical studies

**Nicolas Falgayrettes**[1], **Etienne Patoor**[1], **Franck Cleymand**[2], **Yinka Zevering** [3], **Jean-Marc Perone** [3]*

1 CNRS IRL 2958, GT-CNRS, GeorgiaTech Lorraine, Metz, France, 2 Department of Nanomaterials, Electronics, and Living Systems, Institut Jean Lamour, Nancy, France, 3 Department of Ophthalmology, Metz-Thionville Regional Hospital Center, Lorraine University, Mercy Hospital, Metz, France

* jm.perone@chr-metz-thionville.fr

## Abstract

### Background

The steep cornea in keratoconus can greatly impair eyesight. The etiology of keratoconus remains unclear but early injury that weakens the corneal stromal architecture has been implicated. To explore keratoconus mechanics, we conducted two numerical simulation studies.

### Methods

A finite-element model describing the five corneal layers and the heterogeneous mechanical behaviors of the ground substance and lamellar collagen-fiber architecture in the anterior and posterior stroma was developed using the Holzapfel-Gasser-Ogden constitutive model. The geometry was from a healthy subject. Its stroma was divided into anterior, middle, and posterior layers to assess the effect of changing regional mechanical parameters on corneal displacement and maximum principal stress under intraocular pressure. Specifically, the effect of softening an inferocentral corneal button, the collagen-based tissues throughout the whole cornea, or specific stromal layers in the button was examined. The effect of simply disorganizing the orthogonally-oriented posterior stromal fibers in the button was also assessed. The healthy cornea was also subjected to eye rubbing-like loading to identify the corneal layer(s) that experienced the most tensional stress.

### Results

Conical deformation and corneal thinning emerged when the corneal button or the mid-posterior stroma of the button underwent gradual softening or when the collagen fibers in the mid-posterior stroma of the button were dispersed. Softening the anterior layers of the button or the whole cornea did not evoke conical deformation. Button softening greatly increased and disrupted the stress on Bowman's membrane while mid-posterior stromal softening increased stress in the anterior layers. Eye rubbing profoundly stressed the deep posterior stroma while other layers were negligibly affected.

**Data Availability Statement:** This paper describes a modelling study using the ophthalmological data of one of the authors, which are indicated in the manuscript methods section. All information

needed to reproduce our simulations are provided in the paper.

**Funding:** This work is supported by grant n˚17CP-1391-C63 to N.F. from Region Grand Est, France. The funders had no role in study design, data collection and analysis, decision to publish, or preparation of the manuscript.

**Competing interests:** The authors have declared that no competing interests exist.

**Abbreviations:** EDS, Ehlers-Danlos Syndrome; HGO, Holzapfel-Gasser-Ogden; IOP, Intraocular pressure; LASIK, Laser assisted in situ keratomileusis; OCT, Optical coherence tomography; PRK, Photorefractive keratectomy.

## Discussion

These observations suggest that keratoconus could be initiated, at least partly, by mechanical instability/damage in the mid-posterior stroma that then imposes stress on the anterior layers. This may explain why subclinical keratoconus is marked by posterior but not anterior elevation on videokeratoscopy.

## Introduction

Of the five anatomical layers of the normal cornea, the stroma accounts for 90% of total corneal thickness [1]. Its keratocytes generate a complex architecture composed of ground substance (non-fibrillar collagen and proteoglycans) and fibrillar collagen type I fibrils that are packed into several hundred belt-like lamellae [1–3]. The lamellae are held in place by cross-links created by the ground substance molecules [2, 4]. The anterior and posterior stroma differ markedly in terms of lamellar architecture (Fig 1A): the anterior lamellae are randomly interwoven, thin, and anchored in Bowman's membrane whereas towards the posterior two-thirds of the stroma, the lamellae increasingly start taking orthogonal orientations in the inferior-superior and naso-temporal directions while adopting a stacked plywood-like organization. When these lamellae approach the four limbal regions, they curve to eventually form a pseudo-annulus. Compared to the anterior lamellae, the posterior lamellae are wider, thicker, less densely packed, and highly organized [3, 5–9]. These architectural differences have biomechanical consequences: the anterior stroma has greater cohesive tensile strength [10] and swells less readily [11], while the posterior stroma causes the cornea to resist intraocular pressure (IOP) and maintain adequate corneal curvature [12, 13]. Overall, this stromal architecture endows the cornea with anisotropic mechanical properties [14].

From an engineering point of view, the cornea can be viewed as a laminated collagen fiber composite within a matrix of polyelectrolytes and water. Its mechanics display the habitual nonlinear elastic behavior of such fiber-reinforced composites, namely, corneal stiffness rises when deformation increases and the collagen fibers in the stroma become stretched. This so-called strain-hardening behavior of the cornea is particularly pronounced outside the physiological IOP range of 15–18 mm Hg and is thought to maintain adequate curvature of the tissue during excessive IOP [15]. Thus, the collagen network in the stroma, rather than the other four corneal layers, largely shapes the solid mechanics of the cornea [16].

Since corneal shape reflects the interplay between corneal mechanical stiffness and IOP, corneal ectasia can appear when the structure is compromised due to removal of anterior corneal material (as occurs in refractive surgery) or collagen fiber disorganization. The latter case characterizes keratoconus, which is a progressive, non-inflammatory, asymmetrical corneal disease that is most commonly diagnosed in the third decade of life [17]. Imaging studies show that in keratoconic corneas, the stromal microstructure is disrupted within an inferocentral keratoconic button but not outside it. Specifically, stromal thickness in the button drops markedly, apparently because of lamellar slippage out of the cone and structural changes to the lamellae. In the anterior stroma, the lamellae appear to exhibit less interlamellar adhesion, lose their interwoven pattern and insert more rarely into Bowman's layer. The posterior lamellae also exhibit thinning, splitting, and distortion along with Vogt's striae, which run anteriorly from the deep posterior stroma: while normal corneas also have these striae, they are more numerous and longer in keratoconus. Other notable changes in keratoconus are thinning and fragmentation of Bowman's membrane, thickening of the epithelial layer to fill the void left by

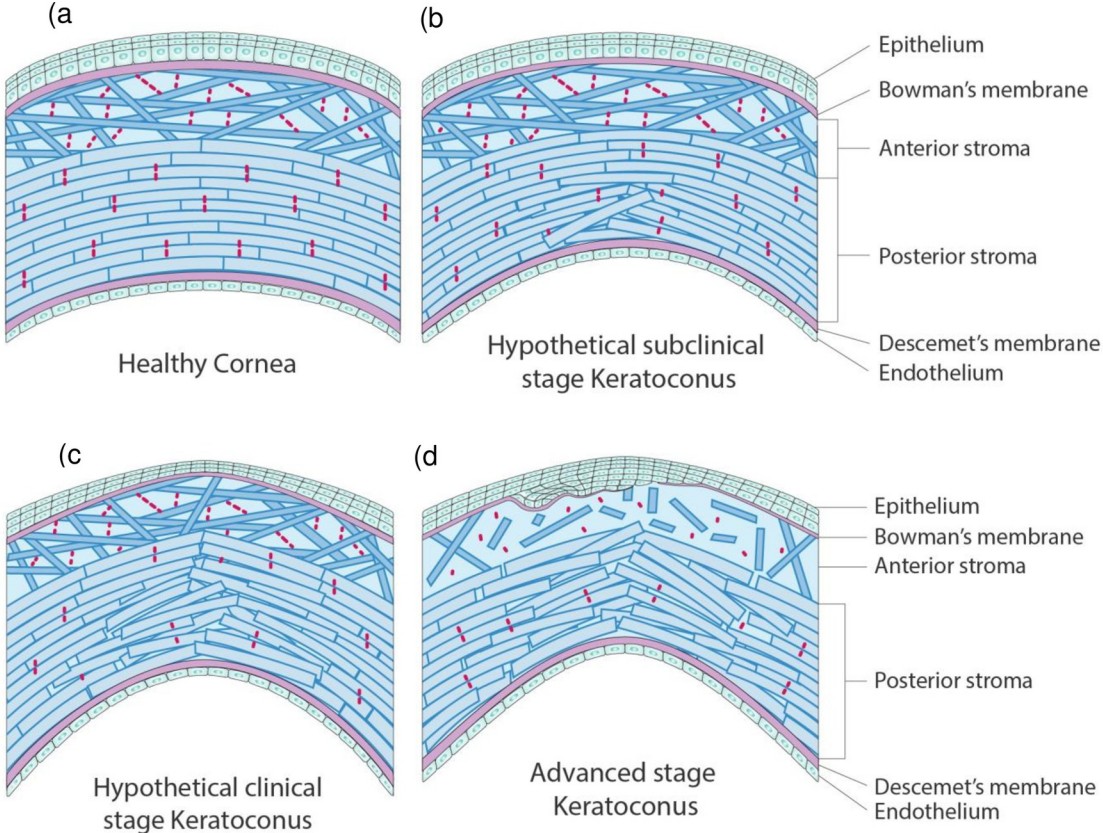

**Fig 1. Schematic depiction of the cornea in cross-section that shows the hypothetical microstructural changes in the stroma that may lead to keratoconus onset and progression.** (a) Healthy cornea. (b) Our hypothesized very early stage keratoconus, which is characterized by a few cross-link breakages in the deep posterior stroma that disrupt the local lamellar organization and induce posterior elevation. (c) An hypothesized later stage in which cross-link breakages have accumulated and spread into the mid stroma. Mid-posterior stromal lamellar disorganization has become more pronounced and epithelial thinning and deformation is now evident. (d) Advanced keratoconus exhibiting critical failure of the stromal microstructure and collagen fiber breaks in Bowman's membrane that result in epithelial outgrowth. Blue bars/bricks = lamellae. Red dashes = interlamellar cross-links generated by ground substance.

the shrunken stroma, and in end-stage disease, rupture of Descemet's membrane and the onset of hydrops [3, 6, 12, 13, 17–23]. These changes are all largely concentrated in the region where the cone forms (Fig 1D).

Mechanical theory on laminated composites [24, 25] suggests that keratoconus is due to repeated excessive stress that results in progressive biomechanical failure of the cornea. Specifically, stress on the cornea could provoke two of the predominant mechanisms by which fiber-reinforced composites fail, namely, fiber breakage and interfiber cross-link breakage. Fiber breakage in keratoconus is evident in Bowman's membrane and Descemet's membrane at later stages of the disease [stages 2 and 5 in the five-stage optical coherence tomography (OCT) classification of keratoconus progression, respectively [18]]. By contrast, cross-link breakage (*i.e.* matrix cracking that leads to fiber/matrix interface debonding) is prevalent in the keratoconic stroma, as evidenced by the disorganization of the stromal lamellae, the localization of the damage, and Vogt's striae. This may reflect the weakness of the stromal cross-links relative to the stiff collagen lamellae in the stroma. Given the potent biomechanical role of the stroma, it seems possible that the accumulation of cross-link breaks in the stroma eventually fatally weakens the cornea and induces its conical IOP-induced elevation [26].

The imaging findings in subclinical keratoconus (forme fruste), the earliest known stage of the disease, suggest that this stromal cross-link breakage starts early in keratoconus pathogenesis. Specifically, the only clinical finding in subclinical keratoconus is posterior elevation on videokeratoscopy: there are no obvious signs of epithelial or Bowman's membrane degradation on slit-lamp/OCT, and the anterior curvature is normal on videokeratoscopy [27] (S1 Fig). Thus, the stroma may develop cross-link breakages that affect the biomechanical properties of the cornea well before significant anterior injuries are detected [28].

The stress that provokes stromal cross-link breakage in keratoconus is not from IOP, which is normal. Rather, it may arise from environmental factors such as eye rubbing: it is generally thought that such factors shape the penetrance of keratoconus-susceptibility genes [29]. It has been hypothesized that the shear stresses invoked by frequent eye rubbing break the interlamellar cross-links either directly [13, 26, 30] and/or indirectly by inducing keratocyte apoptosis, increasing matrix catabolic protein production, and/or impairing collagen/ground substance production [26, 29, 31–33].

The fiber-reinforced structural model of Holzapfel, Gasser, and Ogden (HGO) successfully describes how fibrous biological tissues like the cornea deform when a load (*e.g.* pressure) is applied [34–36]. Multiple studies have shown that when this model is used to describe the fibers throughout the cornea as purely anisotropic (*i.e.* adopting the orthogonal orientation seen in the posterior stroma), it replicates the *in vivo* corneal behavior in response to normal and high IOP, uniaxial tension, incisions for astigmatism, and refractive surgery [37, 38]. Subsequently, on learning about the fiber dispersion gradient in the stroma (*i.e.* changing from anisotropic posteriorly to isotropic anteriorly) [6], several groups adapted their models to reflect this architecture. These models successfully mimic the corneal responses to high IOP, air-puff tonometry, indentation with a probe, eye rubbing, and refractive surgery [5, 7, 16, 39–50]. A number of studies have also used anisotropic stromal models to show that ectatic-like bulging can be achieved by reducing fiber stiffness within part of the cornea [38, 51–53], decreasing local corneal thickness [52], gradually reducing both corneal thickness and material properties [53], or weakening the transversal bonds between collagen fibrils [30]. Moreover, when finite element models were used with keratoconus patient-specific corneal geometries, they accurately replicated the corneal curvature change of the patients after photooxidative collagen cross-linking treatment [53, 54]. A similar approach by Angelillo et al. also demonstrated that ectatic corneas exhibit abnormally high shear stress and disorganized principal stress lines that reflect deviant fiber organization; together, these findings are suggestive of transmission of shear stress onto the stromal matrix in ectasia [55].

However, while models of keratoconus have been very useful for understanding the role of mechanics in corneal ectasia, none have to date successfully achieved the pronounced local bulging that characterizes keratoconus [30, 51–54, 56]. To address this, and particularly to explore the biomechanics of the cornea further, we used the material model developed by Holzapfel, Gasser, and Ogden in 2006 [35] and a corneal geometry that was partitioned according to Meek and Boote [6]; specifically, we split the stroma into anterior, middle, and posterior layers. This approach thus generated a model that mimicked the complex stromal architecture throughout the depth of the cornea. Our approach was inspired by the HGO-based corneal model built by the Pandolfi group [16, 57] and its application by the Ariza-Gracia group in 2015–2016 [45, 46]. We then conducted two numerical studies. First, we investigated which types of weaknesses in specific tissue zones can induce the conical deformation in keratoconus. Second, we applied an eye rubbing simulation to the healthy corneal model to identify the corneal zone that is most likely to be damaged by eye rubbing.

## Methods

### Material model of the stroma

On examining the literature for material models that have been used to study corneal mechanics [5, 7, 16, 39, 41–43, 45–50, 58–60], we found that the HGO model is often employed to describe the stroma, which is the thickest and most mechanically influential layer. This behavior law describes the strain-hardening hyperelasticity and anisotropy of soft tissue that contains several families of fibers with defined direction and dispersion that are embedded within an isotropic matrix [34, 35]. It gives us the ability to describe the complex fiber-reinforced architecture of the corneal stroma and how it deforms and undergoes strain when subjected to external loadings such as IOP [45, 46, 48]. In the HGO model, the deformation is composed of three independent contributions, such that the strain energy density function $W$ relates to the strain invariants $I$ [48] as:

$$W(\bar{I}_1, \bar{I}_2, I_3, \bar{I}_4, \bar{I}_6) = W_{vol}(I_3) + \bar{W}_{iso}(\bar{I}_1, \bar{I}_2) + \bar{W}_{aniso}(\bar{I}_1, \bar{I}_4, \bar{I}_6)$$

where $W_{vol}(I_3)$ accounts for the volumetric response of the material, the isochoric strain-energy function $\bar{W}_{iso}$ describes the isotropic ground substance, and $\bar{W}_{aniso}$ is the isochoric anisotropic contribution that describes two embedded orthogonal collagen fiber families with the same mechanical properties and dispersion; specifically, it addresses the strengthening caused by their stretching at high deformations. The details of these contributions are as follows.

The arguments of the two isochoric contributions are the invariants $\bar{I}_1$, $\bar{I}_2$ of $\bar{C}$, which are expressed as: $\bar{I}_1 = \bar{C}_{II}$, $\bar{I}_2 = \frac{1}{2}\left(\bar{C}_{II}\bar{C}_{JJ} - \bar{C}_{JK}\bar{C}_{KJ}\right)$, with $\bar{C} = \bar{F}^T\bar{F}$ the Cauchy-Green deformation tensor and $\bar{F} = J^{-1/3}F$ is the isochoric deformation gradient. The argument of the volumetric contribution is $I_3 = \det(C)$.

### Volumetric contribution $W_{vol}$

This accounts for the volumetric response of the material. Due to high water content, soft biological tissues are assumed to be largely incompressible and this term is treated as a penalty function enforcing the incompressibility in the form:

$$W_{vol}(I_3) = \frac{1}{D}\left(\frac{I_3^2 - 1}{2} - \ln(I_3)\right)$$

where $\frac{1}{D}$ is a material parameter independent of the deformation. This formulation allows relaxation of the incompressibility condition, which inserts slight unphysical compressibility into the model so that a stable solution is achieved [61]. With the parameter $D = 0.5$, the equivalent Poisson's ratio of the material in linear elasticity is $v = 0.497$.

### Isotropic contribution $\bar{W}_{iso}$

This describes the behavior of the isotropic stromal components, namely, the ground substance and the ~60% of total collagen fibers that are isotropically dispersed. It is modeled according to the isochoric Neo-Hookean strain energy function:

$$\bar{W}_{iso}(\bar{I}_1) = \mu(\bar{I}_1 - 3)$$

where $\mu > 0$ is the shear modulus of the material and describes its mechanical behavior under small deformation. Note that μ, as a measure of isotropic ground-substance stiffness, is an important material parameter that we altered in Numerical Study 1.

## Anisotropic contribution $\bar{W}_{aniso}$

This describes the contribution of two orthogonal collagen fiber families (~40% of total collagen) that bear the same mechanical properties and dispersion. The exponential function addresses fiber strengthening under large deformation:

$$\bar{W}_{aniso}(\bar{I}_1, \bar{I}_4, \bar{I}_6) = \frac{k_1}{2k_2} \sum_{i=4,6} e^{[k_2 \langle \bar{E}_i \rangle^2] - 1}$$

The arguments are the first invariant $\bar{I}_1$ and the two modified invariants $\bar{I}_4$ and $\bar{I}_6$ of $\bar{C}$, which are expressed as: $\bar{I}_1 = \bar{C}_{II}$, $\bar{I}_4 = a_K^{(04)} \bar{C}_{KJ} a_J^{(04)}$, and $\bar{I}_6 = a_K^{(06)} \bar{C}_{KJ} a_J^{(06)}$.

The material constant $k1$ is a fiber stiffness coefficient (MPa) that describes the slope of fiber-stretching behavior. Note that k1, as a measure of collagen-fiber stiffness, is an important material parameter that we altered in Numerical Study 1. The $k2$ constant is a dimensionless parameter used to adjust the stretching level required to engage the fibers. Together they describe fiber-stretching behavior with the basic assumption that collagen fibers can only support tension and buckle under compression.

The strain-like quantity $\bar{E}_i = \kappa(\bar{I}_1 - 3) + (1 - 3\kappa)(\bar{I}_i - 1)$ characterizes fiber family deformation by introducing the anisotropic pseudo-invariants $\bar{I}_i$, which are the squared stretch in the mean direction $a_K^{(0i)}$ for the i-th family of fibers: $\bar{I}_i = a_K^{(0i)} \otimes a_K^{(0i)} : \bar{C}$. In the $\bar{E}_i$ equation, the parameter $\kappa \left( 0 \leq \kappa \leq \frac{1}{3} \right)$ describes the dispersion of fibers around a mean direction. When $\kappa = 0$, the fibers are perfectly aligned (no dispersion). When $\kappa = \frac{1}{3}$, the fibers are randomly distributed and the material becomes isotropic. The model assumes that fibers are dispersed with rotational symmetry around a mean preferred direction $a_K^{(0i)}$ with $\rho(\theta)$ being the orientation density function that characterizes this distribution. Therefore,

$$\kappa = \frac{1}{4} \int_0^\pi \rho(\theta) \sin^3 \theta d\theta$$

describes the degree of anisotropy and represents fiber distribution in an integral sense. For perfectly aligned fibers (*i.e.* $\kappa = 0$), $\bar{E}_i = \bar{I}_i - 1$ with i = 4,6. For randomly distributed fibers $\left( i.e. \ \kappa = \frac{1}{3} \right)$, $\bar{E}_i = \frac{1}{3}(\bar{I}_1 - 3)$. Note that $\kappa$ (referred to as kappa from here on in to distinguish it from k1) is an important material parameter that we altered in Numerical Study 1.

## Modeling the collagen fiber architectural gradient in the stroma

The X-ray studies of the Meek group showed that the collagen fibers in the central and posterior stroma of the cornea follow a distinct orthogonal pattern (one set of fibers each is oriented in the nasal-temporal and superior-inferior directions) and, as they approach the limbus, they start to curve, eventually forming a stiff reinforcing annular ring of fibers around the edge of the cornea in the posterior stroma. Moreover, several studies showed that the anterior lamellae are randomly interwoven and isotropically dispersed whereas in the posterior two-thirds of the stroma, the lamellae start adopting a stacked plywood-like organization [6, 7, 22, 31, 62]. To be able to describe this complex lamellar architecture, we split the stroma into nine different regions that bear distinct lamellar architectures. These regions distinguish the anterior zone from the posterior one and the central area from the peripheral one, with transition zones in between (Fig 2A). We then modeled the deep stroma with a planar distribution of perfectly aligned fibers (kappa = 0, *i.e.* no dispersion) while the anterior stroma was modeled as fully isotropic (kappa = 1/3, *i.e.* complete dispersion). The middle stroma was modeled with intermediate dispersion (kappa = 1/6) (Fig 2A, inset circle). In addition, the stroma was stiffened in the periphery to reflect the presence of

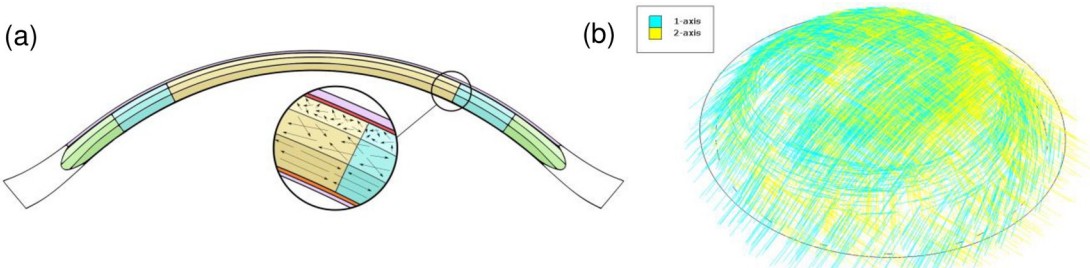

**Fig 2. The model geometry.** (a) Two-dimensional depiction showing the division of the stromal layer of the cornea into three areas that bear their own specific collagen architecture, namely, the central corneal stroma (yellow), the peripheral corneal stroma (green), and the transitional middle area in between (blue). The arrows in the inset figure show the predominant orientations of the lamellae in the anterior stroma (fully dispersed), middle stroma (moderately dispersed), and) posterior stroma (perfectly aligned without any dispersion). In the green zone, the collagen fibers arriving from the corneal center curve to provide annular reinforcement at the corneal edge (not depicted). (b) Three-dimensional depiction of the preferential organization of the collagen lamellae in the posterior stroma, namely, a predominant orthogonal fiber orientation that eventually curves to provide annular reinforcement at the limbus.

reinforcing fibers coming from the sclera and the increasing fiber diameters as they move from the center to the periphery (Fig 2B): this mimics the observations made by Boote et al. [62]. It should be noted that while other groups have achieved a similar gradient of material properties with various mathematical descriptions [5, 7, 16, 39, 41–47, 49, 50, 55], we chose to take our cutting approach so that we could later alter μ, k1, and kappa in specific parts of the corneal stroma in Numerical Study 1 to see if that would reproduce a pathologically weakened structure.

## Material parameters

Since the different dispersion, orientation, and fiber stiffness of the stromal parts impose different "absolute" stiffnesses, we proportionally tuned $\mu$ and $k_1$ in each layer so that the stress distribution throughout the full thickness of the corneal stroma was homogeneous when the corneal model was submitted to normal IOP (this homogeneity is displayed in Fig 5B). To achieve this homogeneity, we conducted inflation tests on a monolayer corneal geometry onto which we imposed the HGO material properties found in the literature for a cornea with fully anisotropic properties to calculate a reference value of apex vertical displacement [45, 46]. The aim was to ensure that a monolayer cornea with fully anisotropic properties such as that in the posterior stroma (no dispersion, kappa = 0) generated the same apex vertical displacement as a cornea with fully isotropic properties such as that in the anterior stroma (full dispersion, kappa = 1/3). We similarly tuned the stiffness parameters of the mildly dispersed layer (kappa = 1/6). These tests together yielded the material parameters of the stroma.

Our review of the literature also showed that the Yeoh hyperelastic model has been used to describe the sclera because it models the strain-hardening behavior of soft biological tissues with an isotropic fiber distribution, which is the case for the sclera. Similar literature researches led us to adopt the Neo-Hookean hyperelastic model to describe the behavior of the epithelium, Descemet membrane, and endothelium, while an isotropic formulation of the HGO law was used to describe the behavior of Bowman's membrane [5, 7, 16, 39, 41–43, 45–50, 58–60]. Table 1 shows the parameters of the HGO, Neo-Hookean, and Yeoh models that were used to describe the indicated layers. Of particular interest are μ (ground substance stiffness), k1 (collagen fiber stiffness), and kappa (fiber dispersion) in the stroma, which we altered in various analyses in Numerical Study 1.

**Table 1. Material properties used to generate a model of the healthy cornea.**

| Parameters (unit: MPa) C10 = μ/2 | | | | | | |
|---|---|---|---|---|---|---|
| **Layer** | **Model** | **Parameters** | | | | |
| **Epithelium** | Neo-Hookean | C10 = 0.002 | D = 17 | | | |
| **Bowman's membrane** | HGO | C10 = 0.09 | D = 0.5 | K1 = 0.9 | K2 = 150 | Kappa = 0.3333 |
| **Descemet—Endothelium** | Neo-Hookean | C10 = 0.01 | D = 4.8 | | | |
| **Sclera** | Yeoh | C10 = 0.085 | C20 = 0.0056 | C30 = 0.23 | D1 = 1 | D2 = D3 = 1 |
| **Stroma center anterior** | HGO | C10 = 0.06 | D = 0.5 | K1 = 0.6 | K2 = 150 | Kappa = 0.333 |
| **Stroma center middle** | HGO | C10 = 0.02 | D = 0.5 | K1 = 0.3 | K2 = 150 | Kappa = 0.16 |
| **Stroma center posterior** | HGO | C10 = 0.01 | D = 0.5 | K1 = 0.1 | K2 = 150 | Kappa = 0.01 |
| **Stroma between anterior** | HGO | C10 = 0.075 | D = 0.5 | K1 = 0.8 | K2 = 150 | Kappa = 0.333 |
| **Stroma between middle** | HGO | C10 = 0.025 | D = 0.5 | K1 = 0.35 | K2 = 150 | Kappa = 0.16 |
| **Stroma between posterior** | HGO | C10 = 0.0125 | D = 0.5 | K1 = 0.11 | K2 = 150 | Kappa = 0.01 |
| **Stroma peripheral anterior** | HGO | C10 = 0.09 | D = 0.5 | K1 = 1 | K2 = 150 | Kappa = 0.333 |
| **Stroma peripheral middle** | HGO | C10 = 0.03 | D = 0.5 | K1 = 0.4 | K2 = 150 | Kappa = 0.16 |
| **Stroma peripheral posterior** | HGO | C10 = 0.015 | D = 0.5 | K1 = 0.12 | K2 = 150 | Kappa = 0.01 |

HGO, Holzapfel-Gasser-Ogden constitutive law.

## Geometrical model and boundary conditions

The corneal dimensions used to construct our geometrical model were those of a healthy young author (NF). They were measured *in vivo* with pachymetry and tomography and are: central corneal thickness = 500 μm, peripheral corneal thickness = 620 μm, corneal height = 3.4 mm, horizontal diameter = 11.8 mm, and vertical diameter = 10.7 mm. The author provided written informed consent to undergo the ophthalmological measurements and this study was approved by the Ethics Committee of the French Society of Ophthalmology (IRB 00008855 Société Française d'Ophtalmologie IRB#1).

The corneal geometry was inserted into a piece of sclera to define smooth boundary conditions and avoid an unphysical cut. Since the cornea was linked to the sclera, the deformations were naturally transmitted in the 6 degrees of freedom. The sclera border was defined as fully blocked and the IOP loading was applied onto the internal surface of the cornea (S2 Fig). The IOP used was that of the author whose corneal dimensions were used for the geometrical model (NF) (17.1 mmHg/2280 Pa).

We meshed the finite element model with 10-node quadratic tetrahedron elements (C3D10) and combined the material properties described above with the stroma-partitioned geometry (Fig 2A). Since all the previous operations were conducted on the *in vivo* IOP-deformed geometry, an inverse analysis working on the corneal geometry (nodes coordinates) was then run to determine the stress-free reference corneal dimensions (*i.e.* when the *in vivo* cornea was not subjected to IOP) with these particular boundary conditions and material properties. For this, we referred to Pandolfi *et al.* [57]. The stop criterion was reached when the reference geometry submitted to IOP matched the *in vivo* geometry previously recreated from the ophthalmological measurements. During this process, the solution was always computed with an implicit scheme in static analysis. The stress-free dimensions were: central corneal thickness = 550 μm, peripheral corneal thickness = 720 μm, corneal height = 2.9 mm, horizontal diameter = 11.8 mm, and vertical diameter = 10.7 mm. The stress-free thicknesses of the epithelium, Bowman's membrane, anterior, middle, and posterior stroma, and Descemet's membrane plus endothelium in the center were 50, 10, 115, 166, 191, and 18 μm, respectively.

On this final model describing the five layers of the cornea in reference configuration and their material properties, we conducted a mesh convergence study to ensure that our mesh was fine enough to obtain accurate solutions and would not demand excessive computing resources. We conducted it by checking two variables that play key roles when the cornea is subjected to IOP, namely, vertical apex displacement and stress in the central anterior stroma (S3 Fig). The solution reached convergence with a mesh of ≥40,000 elements.

### Simulation of eye rubbing

The same corneal model described above was used in Abaqus to simulate contact of the finger with the eyelid over the cornea in Numerical Study 2. For this, we defined a contact pair with the finger as the master surface and the cornea as the slave surface in a surface-based contact model. The tangential behavior between this contact pair was set to be frictionless, meaning that when nodes are in contact they slide on each other without inducing shear forces. This is justified by the presence of the eyelid sliding between the finger and the cornea during eye rubbing. Since we did not want any penetration between this contact pair, we defined normal behavior as a "hard" contact pressure-overclosure formulation in Abaqus. As a constraint enforcement method for hard contact, we started with a direct method, which avoided approximations such as those in penalty or augmented Lagrangian constraint enforcement. Since our simulation converged with this method, we did not have to resort to other methods.

## Results

### Numerical study 1: Identification of the corneal tissue zones whose softening yielded keratoconus

The corneal elevation in keratoconus most often occurs in an inferocentral corneal button [32], which is softer and exhibits lamellar disorganization [6] compared to the surrounding normal tissue [63]. We therefore first assessed whether gradually softening the collagen-based layers (*i.e.* Bowman's layer and stroma) in an inferocentral button in the corneal model geometry would generate keratoconus. For this, we added three concentric sections to the geometry, thus creating a button with a bullseye at the inferocentral cornea (Fig 3A and 3B). Bowman's membrane and all stromal layers in the center, middle, and periphery of the button were then softened by dividing both their $\mu$ (ground-substance stiffness) and k1 (collagen-fiber stiffness) by 30, 20, and 10, respectively. This approach reduces stiffness in a linear pattern, as is typically seen in keratoconus [30]. Normal IOP was then imposed. Softening the button effectively generated a keratoconic phenotype: the cornea bulged in an asymmetrical localized manner with a maximal vertical displacement of 846 μm (Fig 3D). By contrast, the normal cornea adopted a uniform curvature and maximal vertical displacement (U2) of 395 μm (Fig 3C).

Fig 4 graphically summarizes the effect of the 30-20-10 button softening and our subsequent explorations (*i.e.* different levels of softening or softening in specific corneal layers, see Analyses 1–3 below) on IOP-induced anterior and posterior corneal elevation at the corneal and button apices (orange/yellow and light/dark blue bars, respectively). The healthy (unsoftened) cornea is designated Case 1 while the 30-20-10 button softening condition is denoted Case 5. Notably, Fig 4 also shows that the keratoconic bulging in Case 5 (indicated by *) was accompanied by increased thinning of the cornea (green bars in Fig 4): thus, imposition of normal IOP led the normal cornea to thin by 34 μm (from 550 to 516 μm) whereas the cornea with the weakened button thinned by 97 mm (from 550 to 453 μm) (Fig 3C and 3D). The respective thicknesses of 516 and 453 μm for the healthy and keratoconic corneas are close to

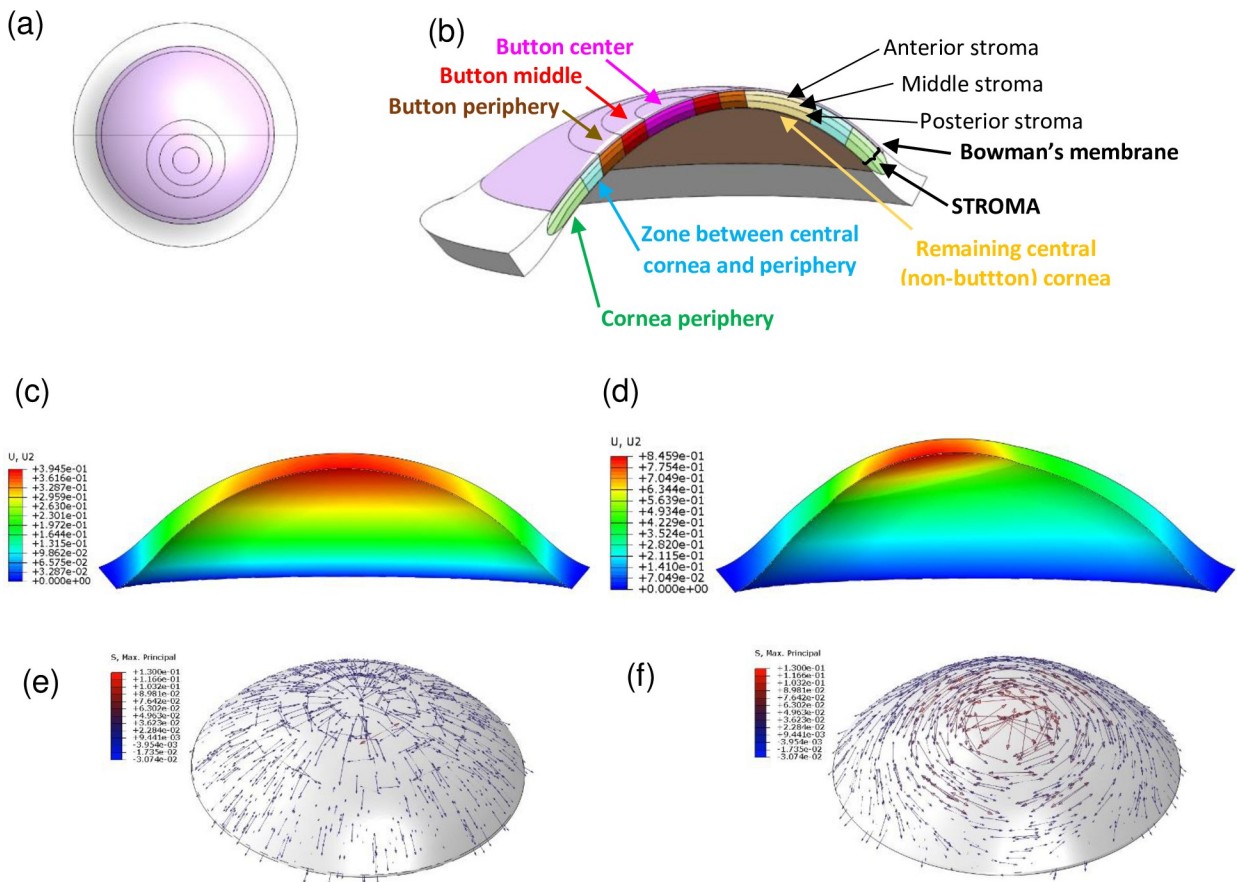

**Fig 3. Generation of a softened button at the inferocentral zone of the model cornea (a, b) and its effect on vertical corneal displacement (U, U2, mm) (c, d) and stress (MPa) on Bowman's membrane (e, f).** (a, b) New subdivisions were added to the model geometry shown in Fig 2A to create a button in the inferocentral area. The button is shown from the anterior view (a) and in cross-section (b). It is composed of three concentric zones, namely, the button center (dark-pink in b), middle (red), and periphery (brown). The yellow tissue is the remaining (non-button) central cornea while the green tissue represents the peripheral cornea, which displays different stromal collagen behavior (curving around the corneal edge as an annular reinforcement). The blue tissue is the transition area between the central and peripheral cornea. Bowman's membrane is shown in light purple. The epithelial layer, Descemet's membrane, and the endothelial cell layer were included in the cornea model but are not shown here. (c–f) Bowman's membrane and all stromal layers in the button were softened in a gradient so that the button center was the softest of the three concentric zones. To achieve this, $\mu$ (ground-substance stiffness) and k1 (collagen-fiber stiffness) of all layers in the button center (pink in b) were divided by 30. This was repeated for the button middle (red in b) and periphery (brown in b) but with increasingly smaller divisors, namely, 20 and 10, respectively. The cornea bearing this 30-20-10-softened button (d, f), and the unsoftened healthy cornea (c, e), were then subjected to normal intraocular pressure. (c–d) The vertical corneal displacement of the corneas is shown in mm. The red zones demonstrate the greatest displacement and the blue zones the least. (e–f) Directions and intensity of maximum principal tensional stress (MPa) on Bowman's membrane. S, Max. Principal, maximal principal stress.

average minimum corneal thicknesses in the literature, namely, 537 μm in normal corneas and 436 (range 297–494) μm in keratoconus [64].

The keratoconic changes in Case 5 also associated with redistribution of the principal stress (which represents maximum tensional stress) on Bowman's membrane. Thus, in the healthy cornea, the tension spread evenly over the meridians and was regularly punctuated by concentric reinforcement stresses (Fig 3E). By contrast, the keratoconic cornea exhibited a completely disrupted state of stress: tensional stress ran predominantly in latitudinal directions and was strong and highly disorganized at the inferocentral apex (Fig 3F). It is notable that simply changing the stiffness of a button of tissue had such a radical effect on the tension throughout Bowman's membrane.

Thus, our button model replicated the pathological corneal deformation and thinning that occurs in keratoconus. To further explore the mechanical stromal architecture-related mechanisms that underlie this disease, we next created four variant models: (1) we examined the effect of milder button softening on corneal shape and thickness as a model of disease progression; (2) we softened the entire cornea to determine whether a purely genetic condition can create conical deformation; (3) we softened specific corneal layers inside the button to determine how much mechanical support each layer contributes to cornea curvature; and (4) we increased the dispersion of the posterior stromal lamellae in the button to determine whether simply altering this property could lead to keratoconus-like elevation.

### Analysis 1: Effect of progressive local softening on corneal shape and thickness

In the model displayed in Fig 3D, keratoconus-like bulging and thinning was observed when we divided both $\mu$ and k1 of the stromal and Bowman's layers in the center, middle, and periphery of the button by 30, 20, and 10, respectively (Case 5 in Fig 4). We then examined the effect of mild and moderate inferocentral button softening on vertical displacement and

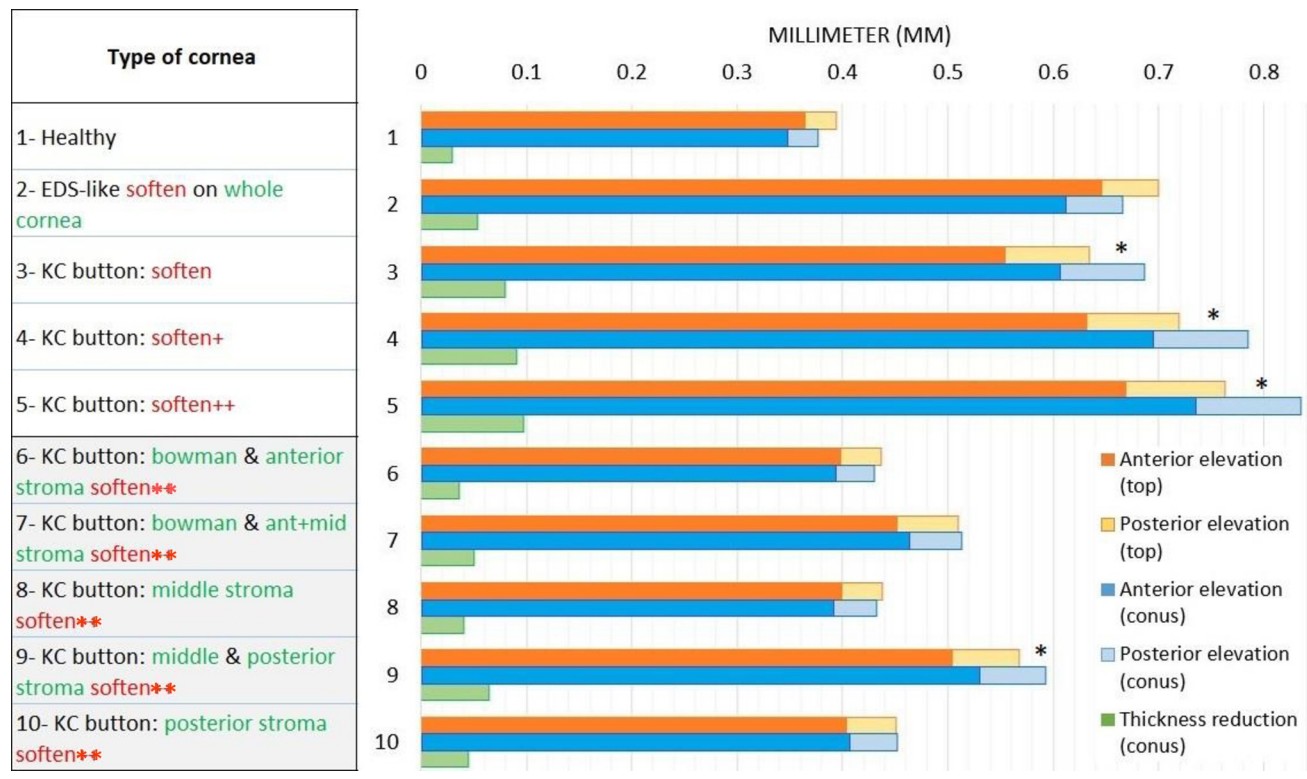

**Fig 4. Graph summarizing the effect of softening specific corneal areas on corneal elevation and thickness.** Summary of the effect of softening the whole cornea (Case 2), a button of corneal tissue (Cases 3–5), or specific corneal layers in the button (Cases 6–10) on the anterior and posterior elevation at the corneal apex (orange and yellow bars, respectively), the anterior and posterior elevation at the button apex (dark and light blue bars, respectively), and loss of corneal thickness (green bars). Cases where the blue bars are higher than the yellow/orange bars indicate keratoconus-like elevation (indicated by *). Note that button elevation is matched by corneal thinning (green bars). Case 1 is the healthy cornea. In Case 2, the whole cornea was softened by dividing $\mu$ (ground-substance stiffness) and k1 (collagen-fiber stiffness) throughout the cornea by 3. In Case 3, an inferocentral button of the cornea was gradually softened by dividing both $\mu$ and k1 in the central, middle, and peripheral button layers (Bowman's membrane and anterior, middle, and posterior stroma) by 10, 6.7, and 3.3, respectively. In Cases 4 and 5, button softening was respectively doubled (to 20, 13.3, and 6.7, respectively; + indicates moderately increased softening relative to Case 3) and tripled (to 30, 20, and 10, respectively; ++ indicates greatly increased softening relative to Case 3). In Cases 6–10, the indicated button layer(s) were softened by dividing their $\mu$ and k1 values in the button center, middle, and periphery by 30, 20, and 10, respectively (** shows that the indicated layers were softened to the same degree). EDS, Ehlers-Danlos Syndrome-like; KC, keratoconus.

thinning as a model of keratoconus progression. Thus, $\mu$ and k1 of these central, middle, and peripheral corneal button regions were instead respectively divided by 10, 6.7, and 3.3 (mild softening) or 20, 13.3, and 6.7 (moderate softening). Indeed, progressive weakening in the button induced increasing corneal steepening: thus, maximal displacement was 395 μm for the healthy cornea (Case 1 in Fig 4) but 682 μm for the mildly weakened button (Case 3 in Fig 4), 781 μm for the moderately softened button (Case 4 in Fig 4), and 846 μm for the 30-20-10 strongly softened button (Case 5 in Fig 4). These changes were paralleled by increasing IOP-induced corneal thinning, namely, from 34 μm in the healthy cornea to 80, 90, and 97 μm, respectively (Cases 3–5 in Fig 4). The close relationship between vertical button displacement and corneal thinning can be attributed to Poisson's effect, which is caused by the tissue deformation. This progressive thinning is consistent with clinical observations of keratoconus, namely, the cornea thins progressively from 550 μm in the normal cornea to 460 μm in the advanced keratoconus stage that associates with the most bulging [65].

Fig 5A–5D depicts in cross-section the effect of IOP on the maximal vertical displacement (left-hand images) and strain distribution (right-hand images) of the healthy cornea (Fig 5A and 5B) and the cornea with the mildly softened button (Fig 5C and 5D). Strain distribution is the ratio of change in the current length to the original length: it quantifies tissue deformation. The left-hand images confirm that mild softening in a button induces a conical shape. The right-hand images show that while the normal cornea exhibited homogeneous deformation under IOP (green/blue-green areas in Fig 5B), the mildly softened cornea displayed concentrated deformation in the inferocentral zone (red/orange areas in Fig 5D).

## Analysis 2: Effect of cornea-wide softening on corneal shape and thickness

Keratoconus has genetic links [66], including with congenital connective tissue disorders such as Brittle Cornea Syndrome. This Ehlers-Danlos Syndrome (EDS) subtype is characterized by deranged synthesis and organization of collagen fibers and thin corneas [67]. It is thought that such collagen disorders may render individuals more prone to eye rubbing-induced keratoconus [68].

We used our model to test whether such a systemic condition, which would presumably soften the entire corneal microstructure, could spontaneously induce keratoconus. Thus, the $\mu$ and k1 variables of all collagen-based layers in our model were divided by 3. The resulting cornea-wide EDS-like softening did induce corneal deformation: the maximal displacement of the EDS-like cornea was 701 μm (Case 2 in Fig 4; also Fig 5E) *versus* 395 μm for the healthy cornea (Case 1 in Fig 4; also Fig 5A) and there was corneal flattening and slight thinning, which is consistent with clinical observations of EDS [67]. However, a conical shape was not achieved. Moreover, like the normal cornea, the EDS-like cornea exhibited homogeneous deformation (green/blue-green areas in Fig 5B and 5F). Thus, cornea-wide softening does not spontaneously induce keratoconus: local softening in a small area is needed to achieve this. This has also been observed previously by early studies on keratoconus with models that described anisotropic stiffness only [38, 51–53].

## Analysis 3: Effect of softening specific collagen-based layers on corneal shape and thickness

To assess the mechanical support provided by each layer to the corneal curvature, we softened one or more of the collagenous corneal layers in the button by dividing $\mu$ and k1 in the selected layer in the button center, middle, and periphery by 30, 20, and 10, respectively. Softening Bowman's layer plus the anterior stroma in the button increased maximum displacement from 395 μm in the healthy cornea (Case 1 in Fig 4; also Fig 6A) to 446 μm but it did not thin the

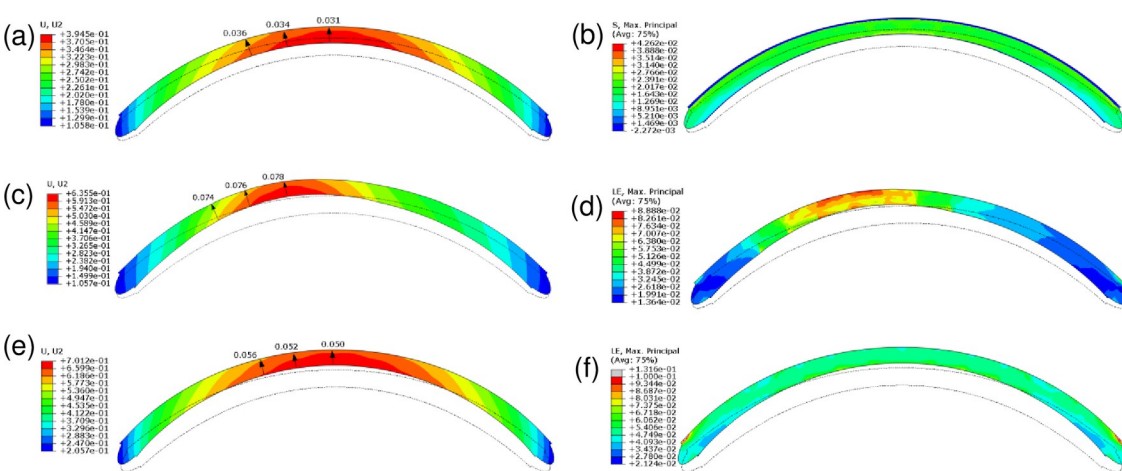

**Fig 5. Effect of global or local softening on corneal vertical displacement (U, U2, mm) (a, c, e) and maximal principal strain (LE, Max.** Principal, MPa) (b, d, f) when the cornea is exposed to normal IOP. (a, b) The healthy cornea. (c, d) The cornea bearing an inferocentral button of tissue that was softened with a third of the softening previously applied in Fig 3 (*i.e. μ* and k1 in Bowman's membrane and the stromal layers in the button center, middle, and periphery were divided by 10, 6.7, and 3.3, respectively). (e, f) An Ehlers-Danlos-like cornea where the whole cornea was softened by dividing *μ* and k1 in Bowman's layer and the stromal layers by 3. Red and blue in (a, c, e) indicate high and low displacement, respectively. Red and blue in (b, d, f) indicate high and low strain, respectively. The position of the cornea without IOP is depicted by the dotted-line shape. The black arrows indicate how much the cornea thins as it deforms under IOP. Avg, average; IOP, intraocular pressure.

cornea or induce any inferocentral bulging (Case 6 in Fig 4; also Fig 6C). It also reduced the stress on these layers while mildly increasing stress on the posterior layers (Fig 6B and 6D). Softening the anterior layers plus the middle stroma increased displacement to 527 μm, created very minor bulging and 1.7-fold more thinning compared to the effect of IOP on the healthy cornea (Case 7 in Fig 4; also Fig 6E). It also further reduced anterior layer stress while imposing more significant stress on the posterior stroma (Fig 6F). Softening only the middle stroma or only the posterior stroma had similar small effects as anterior layer softening (Cases 8 and 10 in Fig 4). However, marked bulging was observed when both the middle and posterior stromal layers were softened: displacement increased to 606 μm (Case 9 in Fig 4; also Fig 6G). This associated with the most pronounced corneal thinning (2.2-fold) as well as increased anterior layer stress (Fig 6H).

### Analysis 4: Effect of disrupting the anisotropic orientation of the middle and/or posterior lamellae on corneal shape

Since we observed that softening the mid-posterior stroma led to the most corneal displacement and thinning (Figs 4, 6G and 6H), we investigated whether simply increasing lamellar disorganization in the posterior stroma of the button would induce corneal elevation. Thus, we increased kappa, which describes collagen fiber dispersion in our model, from 0.01 (no dispersion) to 0.33 (maximal dispersion) in the posterior stroma of the button. Stiffness variables (*μ* and k1) remained unchanged (no softening, see Table 1). Indeed, increasing fiber dispersion in the posterior stroma caused corneal displacement to rise to 461 μm. When we increased the dispersion in the middle stroma as well, this displacement was augmented to 581 μm (Fig 7).

### Numerical Study 2: Effect of eye rubbing on corneal layers

Eye rubbing has been proposed to be an important environmental etiological factor in keratoconus [29]. To test the effect of eye rubbing on the local mechanics of the cornea, we subjected

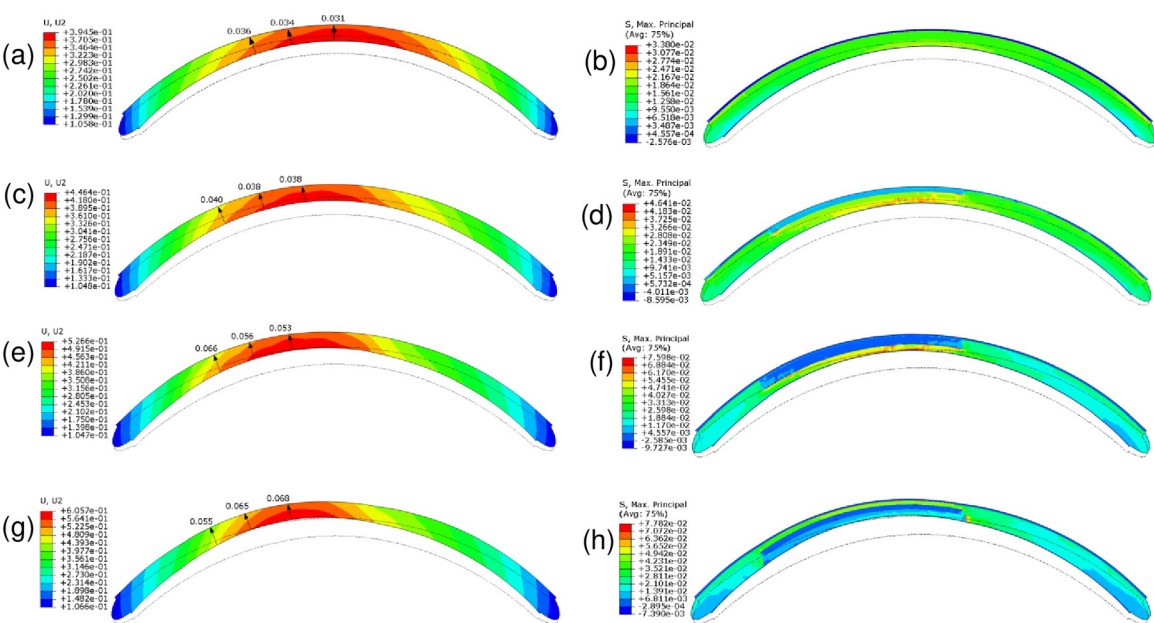

**Fig 6. Effect of softening specific layers in an inferocentral button of tissue on vertical corneal displacement (U, U2, mm) (c, e, g) and maximal principal strain (S.** Max. Principal, MPa) (d, f, h). Gradual softening in the indicated button layer was achieved by dividing its $\mu$ and k1 values in the button center, middle, and periphery by 30, 20, and 10, respectively. (a, b) The healthy cornea. (c, d) Bowman's membrane and the anterior stroma were softened. (e, f) Bowman's membrane and the anterior and middle stromal layers were softened. (g, h) The middle and posterior stromal layers were softened. Red and blue in (a, c, e, g) indicate high and low displacement, respectively. Red and blue in (b, d, f, h) indicate high and low strain, respectively. The position of the cornea without IOP is depicted by the dotted-line shape. The black arrows indicate how much the cornea thins as it deforms under intraocular pressure. Avg, average.

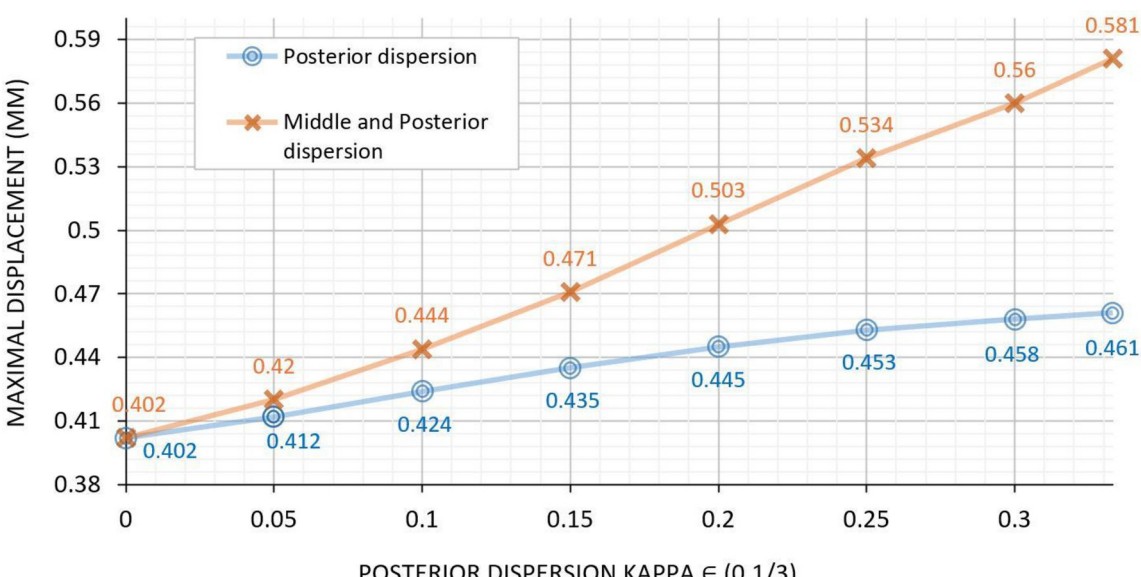

**Fig 7. Effect of increasing the dispersion of the collagen fibers in the posterior stroma (blue plot) and the posterior and middle stroma (orange plot) of the button on corneal displacement (μm).** Only kappa (fiber dispersion) was changed in the selected layer(s). Note that while the x axis shows only the degree of posterior dispersion, we also conducted a second analysis where the fibers in both the middle and posterior stroma were dispersed (orange plot). For this, kappa in the middle stroma was gradually increased from 0.16 to 0.33. (Kappa of 0 = perfectly aligned fibers; Kappa of 1/3 = fully isotropic fibers).

**Fig 8. Depiction of the tension/compression states of the collagen fibers in the anterior and posterior cornea when the cornea is subjected to (a) intraocular pressure alone or (b) external loading such as eye rubbing that bends the cornea inward.**

our healthy cornea model (*i.e.* without any zones of weakening) to two common modes of eye rubbing associated with keratoconus, namely, horizontal rubbing with either a rigid knuckle or a soft finger pad [69]. For this, we respectively moved a rigid and deformable indenter horizontally on the cornea with constant displacement speed. In both simulations, the rubbing was frictionless because it should mimic usual eye rubbing, namely, the rubbing is on the eyelid rather than on the corneal surface. On the basis of a study modeling the biomechanical response to non-contact tonometry [45], we hypothesized that eye rubbing would have two effects. First, it would compress the anterior stromal lamellae and relieve some of the IOP-induced tension on them. Second, rubbing would push the posterior stroma inward, thereby further stretching its lamellae and transferring the stress onto the posterior cornea (Fig 8).

This hypothesis was confirmed by analyzing the first principal stress, which represents maximum tensional stress. Relative to the cornea that was subjected only to IOP, eye rubbing with the knuckle tripled the maximal first principal stress from 0.0338 MPa (without rubbing) to 0.1230 MPa. Importantly, the stress induced by knuckle rubbing was concentrated in the posterior stroma near Descemet's membrane (red/orange regions) while the anterior stroma experienced compressive stress due to the lamellar compression (dark-blue regions) (Fig 9A, 9B and 9E). Fingertip rubbing associated with less stress intensity (0.0627 MPa) but the same stress distribution (Fig 9C–9E).

It has been proposed that the mechanical weakening that mediates keratoconus is due to failure of the cross-links between collagen fibers, which disrupts the lamellar architecture and causes the lamellae to slip out of the inferocentral button [6, 20, 63, 70–72]. This is supported by the numerical analyses of 20 normal and 20 ectatic patient corneas by Angelillo *et al.*, which suggested that keratoconus associates with shear stress on the proteoglycan matrix that holds the collagen lamellae in place [55]. Therefore, we asked whether knuckle rubbing can impose various types of stress on the cornea. The distributions of Cauchy's stresses in the spherical coordinates $\sigma rr$, $\sigma \theta \theta$, and $\sigma r\theta$ represent tensional-opening, shear-sliding, and shear-tearing stress, respectively. Our numerical results show that radial (tensional-opening) stress $\sigma rr$ was an order smaller than the others and mainly composed of compression areas on the anterior layers (Fig 10A). Hoop (shear-sliding) stress $\sigma \theta \theta$ manifested as high tensile posterior stress along with low compressive stress in the anterior layer (-30 kPa) (Fig 10B). This hoop-stress distribution is consistent with findings of other modeling studies examining the effects of indentation induced by air-puff tonometry or finger rubbing [45, 49]. The distribution of shear-tearing stress $\sigma r\theta$ showed shear bands that originated from the posterior stroma and ran to the anterior layers (Fig 10C). The position and orientation of these shear bands is similar to Vogt's striae [19, 21].

## Discussion

Our simulation successfully reproduced the bulging shape of a keratoconic cornea and suggested that the progressive worsening of the disease (as indicated by its bulging and thinning)

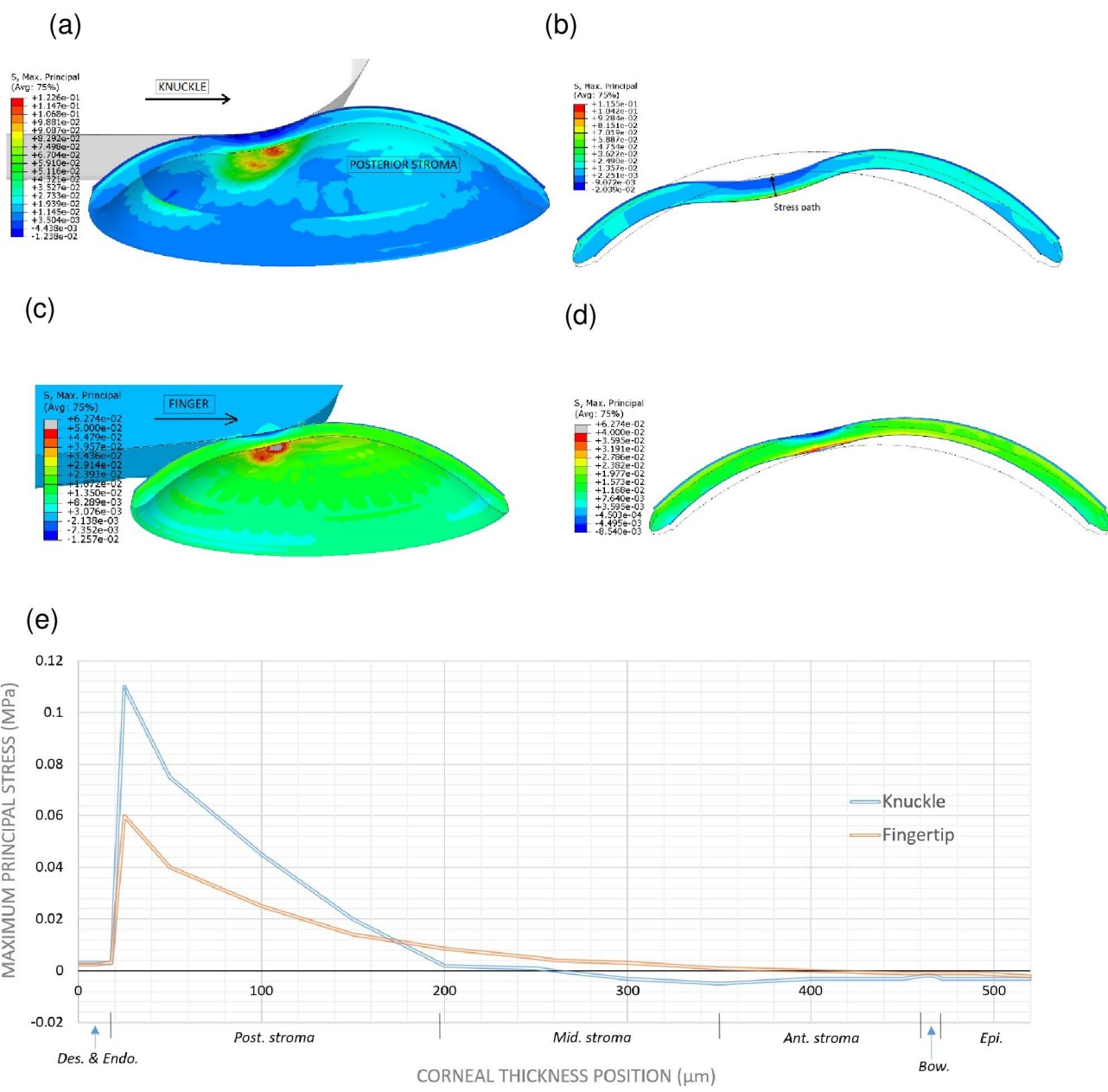

**Fig 9. Effect of knuckle and fingertip rubbing on the maximal tensional stress (MPa) throughout the cornea.** (a, b) The cornea depicted in 3 dimensions (a, c) and 2-dimensional cross-section (b, d) to show the effect of rubbing with a rigid knuckle (a–b) or fingertip (c–d) on stress throughout the cornea. In (a and c), the black arrow indicates the direction of knuckle/finger movement. Descemet's membrane and the endothelium are not displayed in these images. In (b and d), the unstressed cornea is indicated by the fine-line image. In (a–d), red indicates high stress areas. (e) Plot showing the distribution of maximal tensional stress on the various layers of the cornea during knuckle rubbing (blue) and fingertip rubbing (orange). Ant., anterior; Avg, average; Bow, Bowman's layer; Des. & Endo, Descemet's membrane and endothelium; Epi., epithelial layer; Mid., middle; Post., posterior; S, Max Principal, maximum principal strain.

may be due to gradual loss of tissue stiffness that is localized in a concentric area. The study also revealed the importance of the orthogonal architecture of the middle and especially posterior stroma since simply increasing the collagen fiber dispersion in these regions of the corneal button induced subclinical keratoconus-like elevation. Thus, keratoconus may be initiated, at least in part, by localized damage to the deep stromal architecture.

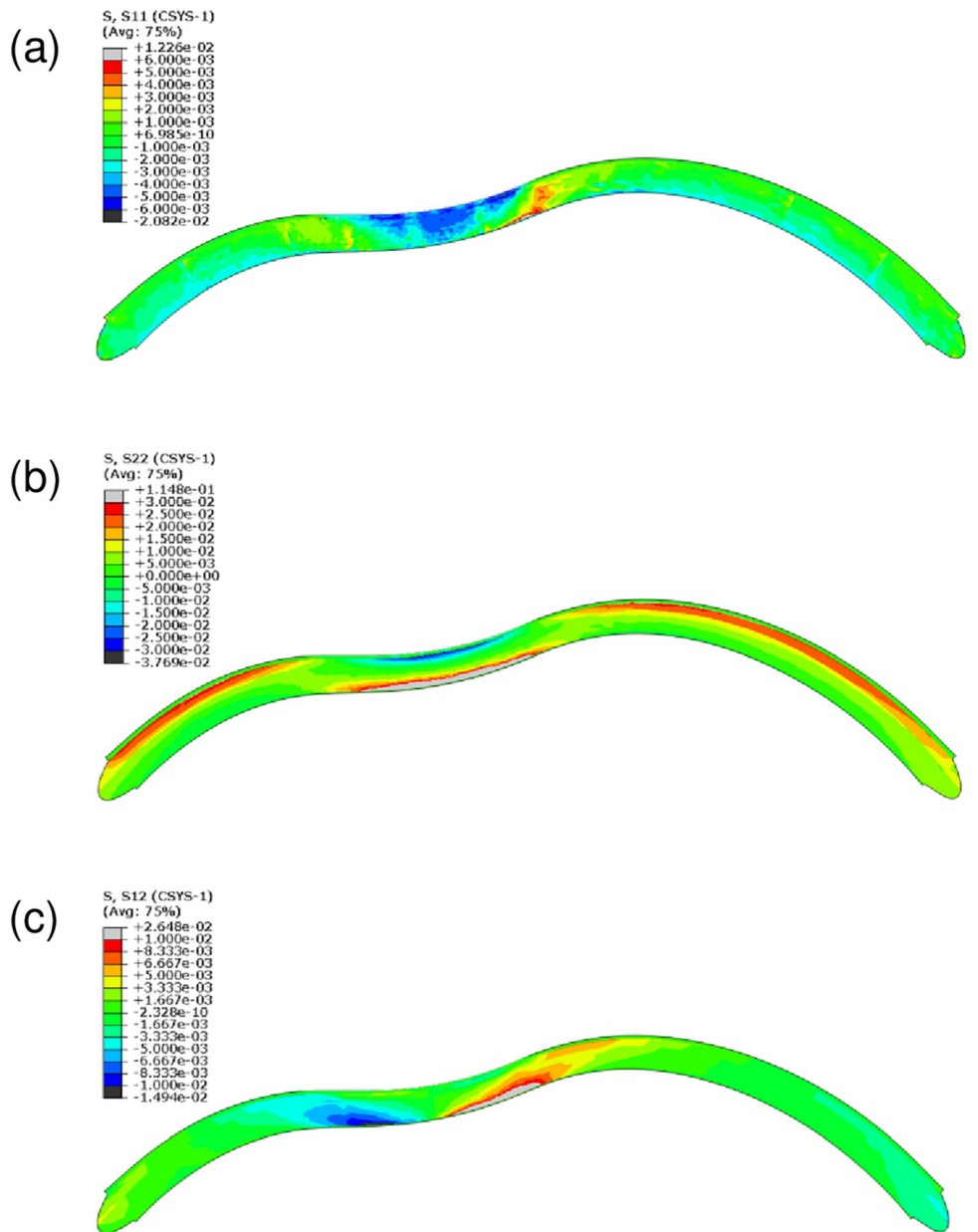

**Fig 10. Two-dimensional cross-section of the cornea showing the distribution of stresses (MPa) from finger rubbing.** (a) Radial stress ($\sigma rr$). This indicates the radial tension towards or away from the central axis of the spherical coordinate system. (b) Hoop stress ($\sigma\theta\theta$). This indicates the circumferential tension in the tangential direction along the cornea. The zone with negative stress (blue) is the compressed anterior zone. The greatest tension is on the most bent zone of the cornea. (c) Shear stress, pushing one part of the structure in one direction and the other in the opposite direction ($\sigma r\theta$). Avg, average.

These findings are consistent with the fact that in the vast majority of cases (99.97%), laser assisted in situ keratomileusis (LASIK) and photorefractive keratectomy (PRK) surgery do not induce corneal bulging [73]. These procedures, which are used to treat refractive errors, involve ablating part of the anterior stroma with or without the epithelium and Bowman's membrane. Thus, the mid-posterior stroma is generally strong enough on its own to resist the IOP and maintain the corneal curvature needed for good refraction.

Our findings are also consistent with the fact that the earliest forms of keratoconus (*i.e.* subclinical or forme fruste keratoconus) demonstrate abnormal posterior elevation on videokeratoscopy despite having normal anterior curvature (S1 Fig) [74]. Moreover, ultrastructural studies by the Akhtar group observed that a corneal button from a patient with 'mild' keratoconus who elected to undergo surgery due to contact lens-related discomfort exhibited normal lamellae throughout the anterior stroma but undulating lamellae in the middle and posterior stroma. Similar studies on more advanced keratoconus cases showed that while they exhibited steeply undulating stromal lamellae throughout the corneal thickness, this disorganization was much more prevalent in the posterior stroma near Descemet's membrane than in the anterior stroma. Thus, early keratoconus associates with lamellar architectural derangement in the posterior stroma that later progresses into the anterior stroma [71, 75].

A widely held view of keratoconus etiopathology proposes that keratoconus is initiated by anterior layer degradation, particularly that of the epithelium. Possible mechanisms could be keratocyte apoptosis, matrix catabolic protein production, and/or impaired collagen/ground substance production [26, 29, 31–33]. This notion is supported by a recent clinical study, which reported that compared to normal corneas, corneas with subclinical keratoconus demonstrate epithelial thinning at the thinnest corneal point [76]. It seems possible that keratoconus is the result of both faulty remodeling of the anterior layers and biomechanical destabilization and cracking in the posterior stroma. One possibility is that posterior damage disrupts the finely tuned mechanical homeostasis of the whole cornea and thereby quickly triggers anterior injury. This is supported by two of our findings: (1) keratoconus-inducing softening of a button in the cornea associated with profoundly deranged and increased tensional stresses on Bowman's membrane (Fig 3E and 3F); and (2) softening the mid-posterior stroma increased the stress in the anterior layers (Fig 6H). Another possibility is that early anterior degradation promotes, or augments, posterior stromal damage. This is supported by our observation that softening the anterior layers of the button significantly increased the stress on the posterior stroma (Fig 6D). Regardless of which possibility is more likely, our findings together with the fact that removal of anterior layers in LASIK or PRK does not induce ectasia in 99.97% of cases suggest that a mechanically weak posterior stroma is needed for the development of ectasia.

## Putative microstructural mechanism by which mid-posterior stromal damage may promote keratoconus

Key clinical features of keratoconus are not only the bulging of the cornea but also its localized thinning. We observed in our simulation that keratoconus-like bulging always associated with proportional thinning of the cornea. Thus, tissue deformation by itself (i.e. without loss of material) can induce corneal thinning. This reflects Poisson's effect, namely, the phenomenon where materials that are expanding in one direction tend to contract in the transverse direction. We propose that in keratoconus, this phenomenon may be induced the loss of interlamellar cross-links and the consequent slipping of lamellae over each other and away from the keratoconic button. This possibility is supported by a growing number of studies and observations [26, 31, 33]. First, photooxidative corneal crosslinking, which introduces crosslinks between collagen fibers and increases corneal stiffness, is an effective treatment for keratoconus [77]. Second, the X-ray scattering study of Meek and Boote on advanced keratoconus buttons showed interlamellar slippage throughout the stroma [6, 63]. Third, Pandolfi et al. showed with a different modeling approach that weakening the transverse bonds between the fibers, but not the fibers themselves, greatly increased corneal deformation [30]. Fourth, older people are less prone to keratoconus than younger people because lysyl oxidation-induced

collagen cross-linking increases with age. Conversely, diabetic patients are protected from keratoconus because of glycosylation-induced cross-linkage [78]. Notably, the modeling study of Studer et al. successfully showed that older patients exhibit more collagen cross-linking than younger patients. Their model employs the same isotropic fiber contribution as in our study but describes the anisotropic fiber contribution in a slightly different way; it also employed a parameter that represents collagen cross-linkage [41]. Fifth, a modeling study by the Pandolfi group that used the geometries of 20 normal corneas and 20 ectatic corneas suggested that ectasia associates with fiber deviation that transfers shear stress onto the ground substance of the stroma [55].

There is also some evidence that the posterior stroma may differ from the anterior layers in terms of susceptibility to cross-link damage. First, the anterior and posterior stroma differ in terms of ground substance: the posterior stroma contains more keratan sulfate, including lumican, more collagen XIII, and less dermatan sulfate [1, 2, 8, 79]. Moreover, the proteoglycans in the posterior stroma are much larger than those in the anterior stroma and resemble sutures; this could shape the disparate swelling properties of the anterior and posterior stroma [80–82]. Second, the posterior stroma contains a weak plane located ~4.5–27.5 μm from Descemet's membrane that allows Descemet's membrane to be easily separated from the anterior layers with a deep air injection [9, 83]. Thus, it is possible that compared to anterior stromal cross-links, those in the posterior stroma are less frequent or more friable.

Thus, we hypothesize that mechanical instability in a small area of the mid-posterior stroma due to loss of cross-links between the collagen lamellae may play an important early role in keratoconus pathogenesis (Fig 1).

## Putative role of eye rubbing in keratoconus-inducing corneal destabilization

Eye rubbing is often quite severe in patients with keratoconus [84] and considerable evidence suggests it may be an etiological factor [85, 86]. Indeed, many ophthalmologists currently recommend their patients to stop rubbing their eyes to avoid disease progression [69, 87]. Since eye rubbing is an external mechanical force, we assessed its effect on our normal corneal model. We observed that eye rubbing tripled the stress on the posterior stroma and caused it to bend inward, therefore tensely stretching the posterior layers. By contrast, the anterior layers, which were in direct/near contact with the knuckle/finger, were subjected to small compressive mechanical stress. This pattern was also replicated by a numerical study employing the Pandolfi model of the healthy cornea [16, 57] that showed the *in vivo* corneal responses to a probe pressed onto the corneal apex or air-puff tonometry [48]. Similarly, the recent numerical mechanobiology study by Pant et al. showed that when modeled keratocytes in the anterior, middle, and posterior stroma were subjected to knuckle-like rubbing, the posterior keratocyte experienced the greatest maximum principal stress whereas the anterior keratocyte experienced negative stress [88]. Our simulation also displayed shear stresses starting from the posterior stroma and running towards the anterior layers. This is evocative of Vogt's striae, a clinical sign often associated with keratoconus. These striae are more numerous in keratoconus than in healthy corneas [89] and may indicate an history of high mechanical stress on the stromal microstructure [19].

The possibility that keratoconus is initiated by external mechanical stress, potentially by eye rubbing, is also supported by the fact that: (1) the structural damage in keratoconus is limited to an inferocentral corneal button, which is also the thinnest central area [6, 32, 63], and we showed that gradually softening such a button of tissue indeed generated keratoconic elevation; (2) the earliest sign of keratoconus is posterior corneal elevation, which is suggestive of damage in this area; and (3) cornea-wide EDS-like softening [67] did not evoke keratoconus.

Thus, it seems possible that eye rubbing induces concentrated mechanical stress in the posterior stroma. The interlamellar links in this region may not be designed to deal with repeated bending stress and buckling and eventually break, causing posterior lamellar disorganization and eventually keratoconus [2, 4, 6, 13, 26, 30, 31, 33]. It is likely that this mechanical effect works in conjunction with other likely etiological contributors, including those operating in the anterior layers [29].

## Study strengths and limitations

This study was based on a well-established model of the cornea that replicates its complex stromal architecture and successfully predicts corneal responses to a variety of insults, including surgery that ablates the anterior layers [16, 43, 44, 47–49, 55, 57]. The novelty of our study lies in the use of a different geometrical approach, namely, the splitting of the stroma into different parts, which allowed us to examine the relative contributions of each part to corneal mechanics and keratoconus pathogenesis. This approach led for the first time to a convincing keratoconic bulge that associated with localized corneal thinning. However, our study is also limited by the fact that the model focuses on mechanical aspects and does not embrace the complexity of the cellular and biochemical mechanisms that participate in corneal physiology and keratoconus pathogenesis, including those that operate in the anterior corneal layers. Modeling studies that include key pathogenic matrix remodeling processes such as collagen turnover, enzymatic degradation, and mechanotransduction are needed to determine the role(s) of these processes in keratoconus. Consequently, our study should only be seen as generating new hypotheses for subsequent experiments.

## Conclusions

This study suggests that the posterior stroma may play an important early role in keratoconus pathogenesis. If validated by other experiments, these findings suggest that characterizing the mechanical stability of the posterior stroma may help to diagnose keratoconus at a very early stage, thereby facilitating early interventions that prevent keratoconus progression.

## Supporting information

**S1 Fig. Two cases of subclinical keratoconus.** (a–d) Videokeratoscopy (a–b) and OCT (c–d) in a patient with diagnosed keratoconus in the left eye (a, c) and suspicion of subclinical keratoconus in the right eye (b, d). Videokeratoscopy showed posterior corneal elevation in the right eye whereas OCT showed no abnormal signs. (e–f) Videokeratoscopy in another patient with bilateral subclinical keratoconus. Visual acuity is preserved and the thickness of the corneas is almost normal but both the left (e) and right (f) eye exhibit posterior elevation. OCT, optical coherence tomography.
(DOCX)

**S2 Fig. Internal pressure and encastre boundary condition defined in the model.**
(DOCX)

**S3 Fig. Mesh convergence study.**
(DOCX)

## Acknowledgments

We thank Alessio Gizzi of Università Campus Bio-Medico, Roma, Italy for his invaluable help when developing the simulation.

## Author Contributions

**Conceptualization:** Nicolas Falgayrettes, Etienne Patoor, Franck Cleymand, Jean-Marc Perone.

**Data curation:** Nicolas Falgayrettes, Etienne Patoor.

**Formal analysis:** Nicolas Falgayrettes.

**Funding acquisition:** Etienne Patoor, Franck Cleymand, Jean-Marc Perone.

**Investigation:** Nicolas Falgayrettes, Etienne Patoor, Franck Cleymand, Jean-Marc Perone.

**Methodology:** Nicolas Falgayrettes, Etienne Patoor, Franck Cleymand, Jean-Marc Perone.

**Project administration:** Nicolas Falgayrettes, Etienne Patoor, Franck Cleymand, Jean-Marc Perone.

**Resources:** Nicolas Falgayrettes, Etienne Patoor, Franck Cleymand, Jean-Marc Perone.

**Software:** Nicolas Falgayrettes, Etienne Patoor, Franck Cleymand, Jean-Marc Perone.

**Supervision:** Nicolas Falgayrettes, Etienne Patoor, Franck Cleymand, Jean-Marc Perone.

**Validation:** Nicolas Falgayrettes, Etienne Patoor, Franck Cleymand, Jean-Marc Perone.

**Visualization:** Nicolas Falgayrettes, Etienne Patoor, Franck Cleymand, Jean-Marc Perone.

**Writing – original draft:** Nicolas Falgayrettes.

**Writing – review & editing:** Nicolas Falgayrettes, Etienne Patoor, Franck Cleymand, Yinka Zevering, Jean-Marc Perone.

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
