## [Decision Letter · Decision Letter 0]

23 Aug 2022

PONE-D-22-17387Biomechanics of keratoconus: two numerical studiesPLOS ONE

Dear Dr. Perone,

Thank you for submitting your manuscript to PLOS ONE. After careful consideration, we feel that it has merit but does not fully meet PLOS ONE’s publication criteria as it currently stands. Therefore, we invite you to submit a revised version of the manuscript that addresses the points raised during the review process. Please, notice that more information about the FEM model is required. Include all the relevant information (mesh convergence model, contact model if used,...)

We look forward to receiving your revised manuscript.

Kind regards,

Antonio Riveiro Rodríguez, PhD

Academic Editor

PLOS ONE

  "We thank Alessio Gizzi of Università Campus Bio-Medico, Roma, Italy for his invaluable help when developing the simulation. This work is supported by grant n°17CP-1391-C63 to N.F. from Region Grand Est, France."

 "This work is supported by grant n°17CP-1391-C63 to N.F. from Region Grand Est, France.

Reviewers' comments:

Reviewer's Responses to Questions

**Comments to the Author**

1. Is the manuscript technically sound, and do the data support the conclusions?

Reviewer #1: Yes

Reviewer #2: Partly

2. Has the statistical analysis been performed appropriately and rigorously? 

Reviewer #1: Yes

Reviewer #2: N/A

3. Have the authors made all data underlying the findings in their manuscript fully available?

Reviewer #1: Yes

Reviewer #2: No

4. Is the manuscript presented in an intelligible fashion and written in standard English?

Reviewer #1: Yes

Reviewer #2: No

5. Review Comments to the Author

Reviewer #1: The paper presents two numerical studies regarding the biomechanics of keratoconus. According to the reviewer’s opinion, the paper is well-structured and clear. The topic is interesting and falls within the aim of the journal. In addition, the results are well-presented and could be helpful to further develop the same topic. Therefore, the paper can be accepted for publication in the current form.

Reviewer #2: The study aims to develop a finite element model for the biomechanics analysis of kerataconus disease. They have used two different models and performed parametric analysis based on idealized geometries. They have related the numerical results to clinical observations. The aim of the study is within the scope of the journal but the manuscript needs revision. I have several suggestions before further assessment. Please see my comments below:

There is almost no information about the finite element model. What kind of parameters are set in Abaqus?

Please share the mesh convergence test results (which might be a figure in the appendix) for finite element analysis. Which parameter was set for comparison?

Did you use any contact model in finite element analysis?

Please provide more technical details about how you performed these calibrations:

"An iterative approach was used to define the material properties in the different corneal layers and

212 sublayers in our healthy cornea model [49,50,62,65]. The final model was calibrated on the basis of

213 a sensitivity analysis to deform under IOP with an apex displacement between 390 and 400 μm

214 [16,30]."

"To reproduce a homogenous stress state under IOP alone, we took the mechanical

226 properties of the different layers from the literature [41,49,50,62,65,66] and calibrated the various

227 parameters as indicated in Table 1."

Please describe the cases before their usage:

"Figure 4 graphically presents the effect of the 30-20-10 button softening (Case 5) on IOP-

260 induced anterior and posterior corneal elevation at the corneal and button apices (orange/yellow and

261 light/dark blue bars, respectively). "

Is this correct? Fingertip rubbing stress is given higher than eye rubbing with the knuckle although the vice versa is stated?

"This hypothesis was confirmed by analyzing the first principal stress, which represents

372 maximum tensional stress. Relative to the cornea that was subject only to IOP, eye rubbing with the

373 knuckle tripled the maximal first principal stress from 0.0338 MPa (without rubbing) to 0.1230 MPa.

374 Moreover, the stress induced by knuckle rubbing was concentrated in the posterior stroma near

375 Descemet’s membrane (red/orange regions) while the anterior stroma experienced negative stress due

376 to the lamellar compression (dark-blue regions) (Fig. 9a, b, e). Fingertip rubbing associated with less

377 stress intensity (0.627 MPa) but the same stress distribution (Fig. 9c–e)."

There seems other zones in the Figure 3?:

"It is composed of three concentric zones that describe the

853 button center (dark-pink), middle (red), and periphery (brown). "

Can you show the zones in Figure 3 with arrows as well?:

"It is composed of three concentric zones that describe the

853 button center (dark-pink), middle (red), and periphery (brown). "

Some figures such as Figure 3, 5 or 6 have missing units.

Figure 7: What is the displacement unit?

Figure 4: What are the meanings of +, ++? Please describe them in the figure caption.

Please check the spelling and English language of the whole manuscript. There are many grammatical and spelling mistakes.

Figures and Tables: Describe all of the abbreviations used in the figures/tables including supplementary ones in the corresponding caption.

There are many abbreviations. So, please add a table of acronyms.

6. PLOS authors have the option to publish the peer review history of their article (what does this mean?). If published, this will include your full peer review and any attached files.

Reviewer #1: No

Reviewer #2: **Yes: **Senol Piskin

---

## [Author Response · Author response to Decision Letter 0]

14 Oct 2022

Point-by-point responses to editor and reviewer comments

Responses to the editor

Please, notice that more information about the FEM model is required. Include all the relevant information (mesh convergence model, contact model if used,...)

Reply: We have provided this information as detailed below in our replies to Reviewer 2.

Reply: We have ensured that the paper meets PLOS One style requirements. We also ensured that the figures meet PLOS One requirements by using the PACE tool.

 "We thank Alessio Gizzi of Università Campus Bio-Medico, Roma, Italy for his invaluable help when developing the simulation. This work is supported by grant n°17CP-1391-C63 to N.F. from Region Grand Est, France."

"This work is supported by grant n°17CP-1391-C63 to N.F. from Region Grand Est, France.

Reply: The funding-related information in the manuscript was removed. We included the amended funding statement in the revised cover letter (Point No. 5).

Reply: Everything that is needed to reproduce the simulations in our study (the methods and the normal corneal dimensions and intraocular pressure of an author, which we used to develop the model) is available in the paper. A comment relating to this (Point No. 6) has been added to the cover letter.

Reply: We have removed the phrase that refers to these data.

Reply: We have added the captions of the Supporting Information to the end of the manuscript. We have also checked to make sure the references are correct.

Responses to the Reviewers

Reviewer #1: 

The paper presents two numerical studies regarding the biomechanics of keratoconus. According to the reviewer’s opinion, the paper is well-structured and clear. The topic is interesting and falls within the aim of the journal. In addition, the results are well-presented and could be helpful to further develop the same topic. Therefore, the paper can be accepted for publication in the current form.

Reply: We are very grateful for the time you spent on our manuscript, your thoughtful consideration, and your positive decision. Thank you very much!

Reviewer #2: 

The study aims to develop a finite element model for the biomechanics analysis of kerataconus disease. They have used two different models and performed parametric analysis based on idealized geometries. They have related the numerical results to clinical observations. The aim of the study is within the scope of the journal but the manuscript needs revision. I have several suggestions before further assessment. 

Reply: Thank you very much for your time, your thoughtful and careful review of our manuscript, and your helpful comments. We have addressed all comments to the best of our ability and feel that this has significantly improved the manuscript.

Please see my comments below:

Q1. There is almost no information about the finite element model. What kind of parameters are set in Abaqus?

Q4. Please provide more technical details about how you performed these calibrations:

"An iterative approach was used to define the material properties in the different corneal layers and

212 sublayers in our healthy cornea model [49,50,62,65]. The final model was calibrated on the basis of

213 a sensitivity analysis to deform under IOP with an apex displacement between 390 and 400 μm

214 [16,30]."

"To reproduce a homogenous stress state under IOP alone, we took the mechanical

226 properties of the different layers from the literature [41,49,50,62,65,66] and calibrated the various

227 parameters as indicated in Table 1."

Q2. Please share the mesh convergence test results (which might be a figure in the appendix) for finite element analysis. Which parameter was set for comparison?

Reply: We apologize for the lack of methodological detail. We have placed Comments Q1, Q4, and Q2 together so that we can explain our modeling approach more cohesively.

The parameters we used are shown in Table 1. Of particular interest are µ (ground substance stiffness) and k1 (collagen fiber stiffness), which we varied in Numerical Study #1 (Analyses 1–3) to assess the effect of softening specific regions of the cornea on corneal elevation and thinning. Another important parameter is kappa, which describes the dispersion of fibers around a mean direction and changes in the different parts of the stroma. In Analysis 4 of Numerical Study #1, we increased kappa in the middle and/or posterior stroma to disperse the fibers, and then examined the effect on corneal elevation. 

Our normal cornea model was constructed as follows: First, we examined the literature for material models that are often used in corneal biomechanics. This showed that the HGO model is often employed to describe the stroma. As indicated in the Introduction, this material model describes the strain-hardening hyperelasticity and anisotropy of soft tissue that contains several families of fibers with defined direction and dispersion that are embedded within an isotropic matrix. Moreover, the Yeoh hyperelastic model has been used to describe the sclera because it describes the strain-hardening behaviour of soft biological tissues that have an isotropic fiber distribution. Similar literature researches led us to adopt the Neo-Hookean hyperelastic model to describe the behavior of the epithelium, Descemet membrane, and endothelium, while an isotropic formulation of the HGO law was used to describe the behavior of Bowman’s membrane. The material parameters of these models are all shown in Table 1. 

Second, we assigned material orientations to the collagen fibers in the stroma to mimic the collagen architecture revealed by the X-ray studies of Meek et al. [ref 6 in our paper]. These studies showed that the collagen fibers in the posterior stroma of the cornea follow a distinct orthogonal pattern (one set of fibers each is oriented in the nasal-temporal and superior-inferior directions) and, as they approach the limbus, they start to curve, eventually forming a stiff reinforcing annular ring of fibers around the edge of the cornea in the posterior stroma (see Figure 2b). This approach has been used in many other studies on corneal mechanics (e.g. refs 16, 45–48, 50, 58). 

Third, we cut the stroma into several parts so that we could assign isotropic fiber dispersion in the anterior stroma and completely anisotropic fiber dispersion in the posterior stroma, with a transition zone in between (Fig. 2a). Others have achieved a similar gradient of fiber dispersion with complex mathematical calculations (e.g. refs 5, 7, 16, 39–41). We chose to take this cutting approach so that we could alter µ, k1, and kappa in specific parts of the corneal stroma. 

Fourth, because the different dispersion, orientation, and fiber stiffness of the stromal parts impose different “absolute” stiffnesses, we proportionally tuned the HGO parameters µ and k_1 so that the stress distribution throughout the full thickness of the corneal stroma was homogeneous when the corneal model was submitted to normal IOP: this homogeneity is displayed in Fig. 5b. To achieve this homogeneity, we conducted an inflation test on a monolayer corneal geometry with the material properties found in the literature for a cornea with fully anisotropic properties (no dispersion) to calculate a reference value of apex vertical displacement. We then tuned the stiffness parameters of layers with different dispersion so that we would achieve the same vertical displacement the monolayer geometry.

Fifth, we meshed the finite element model with 10-node quadratic tetrahedron elements (C3D10) and combined the material properties described above with the stroma-partitioned geometry (Fig. 2a). We applied boundary conditions of IOP and null displacements on the scleral border as shown in S2 Fig. All these operations were conducted on the in vivo IOP-deformed geometry recreated from the ophthalmological measurements of one of the authors.

S2 Fig. Internal pressure and encastre boundary condition defined in the model.

Sixth, to find the stress-free reference geometry of the cornea with these particular boundary conditions and material properties, we ran an inverse procedure working on the corneal geometry (nodes coordinates). The stop criterion was reached when the deformed reference geometry matched the in vivo IOP-deformed geometry that had been recreated from the ophthalmological measurements.

Thus, the final model on which we ran the parametric analyses consisted of the defined material properties assigned to the partitioned stroma and the boundary conditions of loading pressure and null displacements applied to this reference geometry. During this process, the solution was always computed with an implicit scheme in static analysis.

Seventh, we conducted a mesh convergence study using two variables that play key roles when the cornea is subjected to IOP, namely, vertical apex displacement and stress in the central anterior stroma. When we doubled the number of tetrahedral elements at each iteration, the results reached convergence with a mesh of ≥40,000 elements. Given that we had to run a complete parametric analysis with numerous simulations, we therefore used 40,000 elements during the study, since this gave the best balance between precision and speed of calculation (S3 Fig.).

S3 Fig. Mesh convergence study.

To address Q1, Q4, and Q2, we revised the Methods section by:

1. Taking information regarding the computer modeling from Supporting Information and adding it to the Methods. 

2. Completely rewriting the Methods to make the points above (lines 158–291).

3. Adding new references that show the use of the HGO, Yeoh, and Neo-Hookean models to describe the corneal layers (e.g. ref 64).

4. Adding supplementary figures demonstrating our approach (new S2 Fig and new S3 Fig).

Q3. Did you use any contact model in finite element analysis?

Reply: The same corneal model with fiber reinforcement described above was used in Abacus to simulate contact of the finger with the eyelid over the cornea. For this, we defined a contact pair with the finger as the master surface and the cornea as the slave surface in a surface-based contact model. The tangential behavior between this contact pair was set to be frictionless, meaning that when nodes are in contact they slide on each other without inducing shear forces. This is justified by the presence of the eyelid sliding between the finger and the cornea during eye rubbing. Since we did not want any penetration between this contact pair, we defined normal behavior as a "hard" contact pressure-overclosure formulation in Abaqus. As a constraint enforcement method for hard contact, we started with a direct method, which avoided approximations such as those in penalty or augmented Lagrangian constraint enforcement. Since our simulation converged with this method, we did not have to resort to other methods.

To address this point, we added a new section to the Methods (lines 292–303).

Q5. Please describe the cases before their usage:

"Figure 4 graphically presents the effect of the 30-20-10 button softening (Case 5) on IOP-

260 induced anterior and posterior corneal elevation at the corneal and button apices (orange/yellow and

261 light/dark blue bars, respectively). "

Reply: To address this comment, we changed the text as follows:

“Fig 4 graphically summarizes the effect of the 30-20-10 button softening and our subsequent explorations (i.e. different levels of softening or softening in specific corneal layers, see Analyses 1–3 below) on IOP-induced anterior and posterior corneal elevation at the corneal and button apices (orange/yellow and light/dark blue bars, respectively). The healthy (unsoftened) cornea is designated Case 1 while the 30-20-10 button softening condition is denoted Case 5.” (lines 320–324)

Q6. Is this correct? Fingertip rubbing stress is given higher than eye rubbing with the knuckle although the vice versa is stated?

"This hypothesis was confirmed by analyzing the first principal stress, which represents

372 maximum tensional stress. Relative to the cornea that was subject only to IOP, eye rubbing with the

373 knuckle tripled the maximal first principal stress from 0.0338 MPa (without rubbing) to 0.1230 MPa.

374 Moreover, the stress induced by knuckle rubbing was concentrated in the posterior stroma near

375 Descemet’s membrane (red/orange regions) while the anterior stroma experienced negative stress due

376 to the lamellar compression (dark-blue regions) (Fig. 9a, b, e). Fingertip rubbing associated with less

377 stress intensity (0.627 MPa) but the same stress distribution (Fig. 9c–e)."

Reply: Thank you very much for picking this up: there was a missing zero (0.627 Mpa should have been 0.0627 Mpa, i.e. double the maximal first principal stress without any rubbing). We put the missing zero in the text. (line 440) 

Q7. There seems other zones in the Figure 3?:

"It is composed of three concentric zones that describe the

853 button center (dark-pink), middle (red), and periphery (brown). "

Can you show the zones in Figure 3 with arrows as well?:

Reply: Yes, we did not describe this figure sufficiently. Briefly, Figure 3 depicts the creation of a bullseye-like inferocentral button on the corneal geometry shown in Figure 2a. This means that the yellow, blue, and green zones shown in Figure 2a are also shown in Figure 3. Figure 3 also depicts the internal concentric zones of the button (pink, red, and brown). 

To make this clearer and Figures 2 and 3 more harmonious, we rewrote the Fig 3 legend (lines 921–941) and modified the Fig 2 legend (lines 911–919). As suggested, we also added arrows and labels to Fig 3b to make the figure clearer.

Q8. Some figures such as Figure 3, 5 or 6 have missing units.

Reply: The missing unit is mm (displacement) or Mpa (strain) in all cases. This is now made clear in the legends to Fig 3 (line 922), Fig 5 (line 963–964), Fig 9 (line 999), and Fig 10 (line 1012). 

Q9. Figure 7: What is the displacement unit?

Reply: The displacement is given as µm in the y-axis title and was added to the Figure 7 legend (line 989).

Q10. Figure 4: What are the meanings of +, ++? Please describe them in the figure caption.

Reply: We added explanations to the Fig 4 legend as follows:

“In Cases 4 and 5, button softening was respectively doubled (to 20, 13.3, and 6.7, respectively; + indicates moderately increased softening relative to Case 3) and tripled (to 30, 20, and 10, respectively; ++ indicates greatly increased softening relative to Case 3). In Cases 6–10, the indicated button layer(s) were softened by dividing their μ and k1 values in the button center, middle, and periphery by 30, 20, and 10, respectively (++ indicates all layers were softened to the same degree).” (lines 955–961)

Q11. Please check the spelling and English language of the whole manuscript. There are many grammatical and spelling mistakes.

Reply: One of us (YZ) is a medical writer with 25 years experience. She agrees that there were quite a few errors and has read through the manuscript very carefully to correct them. Thank you very much!

Q12. Figures and Tables: Describe all of the abbreviations used in the figures/tables including supplementary ones in the corresponding caption.

Reply: We made sure all abbreviations are spelled out in the figure legends, including the supplementary figures.

Q13. There are many abbreviations. So, please add a table of acronyms.

Reply: A table of acronyms was added (line 24).

---

## [Decision Letter · Decision Letter 1]

6 Nov 2022

PONE-D-22-17387R1Biomechanics of keratoconus: two numerical studiesPLOS ONE

Dear Dr. Perone,

Thank you for submitting your manuscript to PLOS ONE. After careful consideration, we feel that it has merit but does not fully meet PLOS ONE’s publication criteria as it currently stands. Therefore, we invite you to submit a revised version of the manuscript that addresses the points raised during the review process.

Please, address the minor comments provided by Reviewer 2. 

We look forward to receiving your revised manuscript.

Kind regards,

Antonio Riveiro Rodríguez, PhD

Academic Editor

PLOS ONE

Journal Requirements:

Reviewers' comments:

Reviewer's Responses to Questions

**Comments to the Author**

1. If the authors have adequately addressed your comments raised in a previous round of review and you feel that this manuscript is now acceptable for publication, you may indicate that here to bypass the “Comments to the Author” section, enter your conflict of interest statement in the “Confidential to Editor” section, and submit your "Accept" recommendation.

Reviewer #1: All comments have been addressed

Reviewer #2: All comments have been addressed

2. Is the manuscript technically sound, and do the data support the conclusions?

Reviewer #1: Yes

Reviewer #2: Yes

3. Has the statistical analysis been performed appropriately and rigorously? 

Reviewer #1: Yes

Reviewer #2: N/A

4. Have the authors made all data underlying the findings in their manuscript fully available?

Reviewer #1: Yes

Reviewer #2: No

5. Is the manuscript presented in an intelligible fashion and written in standard English?

Reviewer #1: Yes

Reviewer #2: Yes

6. Review Comments to the Author

Reviewer #1: The authors improved the paper as requested by the reviewers.

All the questions have been answered and the statements reported are sufficient to justify final publication.

Reviewer #2: The authors have addressed all of my comments. The manuscript has been improved substantially. Thank you for the revisions. I have a few minor comments:

Can you please a quantification to the abstract results section.

'++' symbol has already been used for the "greatly increased softening" in the figure caption. So another symbol such as '+++' can be used here:

"++ indicates all layers were softened

961 to the same degree"

7. PLOS authors have the option to publish the peer review history of their article (what does this mean?). If published, this will include your full peer review and any attached files.

Reviewer #1: No

Reviewer #2: No

---

## [Author Response · Author response to Decision Letter 1]

10 Nov 2022

Point-by-point responses to editor and reviewer comments (second)

Responses to the editor

Please, address the minor comments provided by Reviewer 2. 

Reply: We have addressed the minor comments provided by Reviewer 2 by adapting Fig. 4 (we replaced the + signs with asterisks). The revised Figure was checked by PACE.

Reply: Our reference list has been checked and is now correct: we removed one duplicate reference (refs 61 and 69 were the same) and corrected 11 refs. There were no retracted papers. 

Responses to the Reviewers

Reply: We thank the reviewers again for their time and for helping us improve our manuscript. 

Rev 2:

'++' symbol has already been used for the "greatly increased softening" in the figure caption. So another symbol such as '+++' can be used here:

"++ indicates all layers were softened

961 to the same degree" 

Reply: We decided to use two asterisks and have made the appropriate change to Figure 4 and the Figure 4 legend (Line 960).

---

## [Editor Report · Decision Letter 2]

17 Nov 2022

Biomechanics of keratoconus: two numerical studies

PONE-D-22-17387R2

Dear Dr. Perone,

We’re pleased to inform you that your manuscript has been judged scientifically suitable for publication and will be formally accepted for publication once it meets all outstanding technical requirements.

Kind regards,

Antonio Riveiro Rodríguez, PhD

Academic Editor

PLOS ONE

---

## [Editor Report · Acceptance letter]

23 Dec 2022

PONE-D-22-17387R2 

Biomechanics of keratoconus: two numerical studies 

Dear Dr. Perone:

I'm pleased to inform you that your manuscript has been deemed suitable for publication in PLOS ONE. Congratulations! Your manuscript is now with our production department. 

Kind regards, 

on behalf of

Dr. Antonio Riveiro Rodríguez 

Academic Editor

PLOS ONE